



# Short and Long-term Grounding Zone Dynamics of Amery Ice Shelf, East Antarctica

Yikai Zhu[1,2], Anna E. Hogg[1], Andrew Hooper[1], Benjamin J. Wallis[1]

[1]School of Earth and Environment, University of Leeds, Leeds, LS2 9JT, United Kingdom
[2]Chinese Antarctic Centre of Surveying and Mapping, Wuhan University, Wuhan, 430070, China

*Correspondence to*: Yikai Zhu (Y.Zhu3@leeds.ac.uk)

**Abstract.** The detection of grounding line (GL) positions in Antarctica is crucial for investigating the stability and health of ice sheets and glaciers. In reality the GL position is not fixed and will migrate upstream or downstream in response to varying

tidal states on an hourly to daily timescale, or in response to longer-term ice dynamic change. However, the magnitude of short and long-term GL migration is not well characterised in many parts of Antarctica. In this study, we employ the Differential Range Offset Tracking method to measure the tidal GL migration on the Amery Ice Shelf in East Antarctica. We delineate 32 GL positions for the year 2021, covering 1,172 km of coastline. The results show that GL migration in this region is not solely dictated by tide amplitude but is also significantly influenced by ice velocity and subglacial bed topography, providing new

insights into the GL dynamics of the region. We also observe significant long-term GL retreat in the eastern part of the Amery Ice Shelf relative to the MEaSUREs Antarctic GL derived from 2000 SAR imagery, with the maximum retreat reaching up to 10 km. Our findings underscore the need for continuous, high-resolution GL monitoring around the whole Antarctic coastline, to improve predictive models of ice sheet responses to climate changes and their subsequent impact on global sea-level rise.

## 1 Introduction

Satellite-based observations acquired over the last 40-years have linked the recent mass loss of the Antarctic Ice Sheet to changes in the floating ice shelves (Paolo et al., 2015) that fringe the majority of its fast-flowing regions (Rignot et al., 2013) and critically control the rate of ice discharge into the ocean by buttressing upstream grounded ice (Gudmundsson, 2013; Haseloff and Sergienko, 2018). The grounding line (GL) is an Essential Climate Variable (Bojinski et al., 2014) where glacial ice shifts from being grounded on bedrock to floating on the ocean (Smith, 1991; Vaughan, 1994), and serves as a critical

boundary condition in numerical ice sheet and glacier models that predict future glaciological evolution (Pattyn, 2018; Seroussi et al., 2014). The GL is a highly sensitive indicator of both glaciological and environmental change. Accurate measurements of the GL are essential for monitoring the stability of ice shelves and predicting potential collapse which is vital for accurately assessing the magnitude of future sea level rise (Joughin et al., 2014). Furthermore, tracking change in the GL location can shed light on the interplay between continental ice sheets and oceanic thermal dynamics, especially the impact of warm ocean

water on ice shelf melt and retreat (Jenkins et al., 2018). Ice thinning and rising sea levels can cause GLs to retreat, whilst thickening or falling sea levels can cause them to advance (Friedl et al., 2020). Thus, continuous surveillance and analysis of



GL positions hold significant importance as they support a broader understanding of global climatic patterns and their subsequent impacts on polar environments and global coastal communities (Stocker, 2014).

Recent advances in remote sensing technology have significantly enhanced our capability to monitor change in GL location, by employing methods such as Double Differential Synthetic Aperture Radar Interferometry (DDInSAR) (Goldstein et al., 1993; Rignot et al., 2016), SAR Differential Range Offset Tracking (DROT)  (Joughin et al., 2016; Marsh et al., 2013; Wallis et al., 2024), Repeat Track Laser Altimetry (RTLA) (Fricker et al., 2009; Fricker and Padman, 2006), and Pesudo Crossover Radar Altimetry (Dawson and Bamber, 2017, 2020). Each technique offers unique advantages in terms of measurement

accuracy, spatial and temporal coverage, and sensitivity to vertical and horizontal movement. Typically, the GL is identified by locating the landward limit of ice flexure driven by tidal motion, however, this process is complicated by the influence of short-term sea level variations which may cause GL migration on an hourly to daily timescale. The short-term variation in GL migration is influenced by factors such as ocean tide amplitude, local bed topography, and the physical properties of the ice. Previous studies have used satellite observations to measure the size of this short-term, tidal GL migration which can vary

significantly in different regions, ranging from several hundred meters to a few kilometres  (Chen et al., 2023; Freer et al., 2023; Milillo et al., 2022). The grounding zone (GZ) in this paper is demarcated by a landward boundary at $F_{min}$ and a seaward boundary $F_{max}$ which encompasses the range of GL migration as influenced by tidal motion (Figure S1). Understanding the interplay between short-term tidal influences and long-term ice dynamic changes is critical for accurately measuring GL migration rates, and importantly, for ensuring that temporary variability is not miss-characterised as a more permanent change.

Short-term GL migration provides valuable insight into the immediate response of ice shelves to tidal forces, while long-term migration reflects a broader response to climatic and environmental change. To improve the accuracy of long-term GL migration rate assessments it is essential to improve our understanding of how these short-term, spatially variable tidal movements affect the overall behaviour of ice sheet GLs.

The dynamics of GL migration on the Antarctic Ice Sheet are governed by a complex interplay of geophysical and oceanographic factors. From a long-term perspective, the thermal regime of adjacent ocean waters exerts a primary role, with elevated subglacial temperatures promoting basal melt and consequent GL retreat (Jenkins et al., 2018). Subglacial topography, especially the presence of retrograde slopes, is a pivotal control on the stability and progression of GL migration (Favier et al., 2014). The process of ice shelf buttressing, where floating ice shelves exert a stabilizing backpressure, is crucial; and reductions

in buttressing strength due to ice shelf attrition or collapse are likely to drive accelerated GL retreat (Rignot et al., 2011). Currently, ocean tide variations are considered the primary factor affecting the short-term GL migration. Rising tides increase buoyancy, lifting the ice shelf from the bed and causing the GL to migrate inland (Gudmundsson, 2006; Padman et al., 2018). The GL then returns to its most seaward position at low tide, with a slight time lag attributed to the ice's viscoelastic properties (Reeh et al., 2003). Variations in subglacial water pressure can also temporarily lift the ice, reducing friction and allowing the

GL to move, such as from seasonal surface meltwater input to the ice-bed interface through supra-glacial lake drainage



(McMillan et al., 2007), active subglacial lake drainage (Sundal et al., 2011) or change geothermal heat driven ice melt (Stearns et al., 2008). Beyond the subglacial system other factors influencing short-term, temporary GL variability include change in ocean currents (Jenkins et al., 2010) and atmospheric pressure (Walker et al., 2013). These factors can act independently or in combination, causing the GL to migrate temporarily. There is a lack of suitable data in the historical satellite archive with sufficient spatial and temporal resolution to characterise short-term GL variability, therefore, our understanding of the mechanisms controlling short-term GL migration is limited. New methods and the larger volume of data acquired by operational satellites such as the European Commission and European Space Agency (EC-ESA) Copernicus Sentinel missions are required to understand the internal mechanisms of the short-term GL migration and to better monitor the stability of ice shelves.

In this study, we use the DROT method to measure the temporal and spatial variation in the location of the GL on the Amery Ice Shelf (AmIS). Our research repeatedly maps the migration pattern of the GL across different tidal states in 2021, providing a detailed record of the GZ dynamics on this ice shelf. We assess the correlation between GL migration and geophysical factors such as ice velocity and subglacial bed topography, exploring the interdependent link between these variables. The results provide a refined perspective on the role of tidal action in shaping the transient behaviours of the GZ on AmIS, and we discuss the potential of DROT for accurately measuring the location and area of ice shelf pinning points. By combining the high temporal-resolution GL measurements obtained using the DROT method with established remote sensing techniques, we aim to improve our understanding of Antarctic GZ dynamics which will help improve the accuracy of sea-level rise projections in a warming climate.

## 2 Methods and Data

### 2.1 Study Area

The AmIS is a prominent feature in East Antarctica and is significant for its role in the continent's glacial dynamics. With an area of about 60,000 km$^2$, AmIS is the third largest ice shelf in Antarctica and the largest ice shelf in East Antarctica. AmIS is the terminus for three major glaciers, namely Fisher, Mellor, and Lambert, which form part of the expansive Lambert-Amery system that is responsible for draining approximately 16 % of the East Antarctic Ice Sheet (Galton-Fenzi et al., 2012). The AmIS is distinct from other Antarctic ice shelves due to its long, narrow sub-ice-shelf cavity and its location at the northernmost latitude (69°S) of the Antarctic coastline, which exposes it to different environmental conditions. Unlike ice shelves in the Amundsen Sea and on the Antarctic Peninsula, AmIS has relatively low levels of basal melt, with modest thickening across the majority of its area and thinning at its most inland reaches (Davison et al., 2023), rendering it potentially less susceptible to rapid disintegration.



The AmIS's complex shoreline and GL have been the subject of extensive research for over half a century. Pioneering field surveys in the late 1960s laid the groundwork for our current understanding of the GL (Budd et al., 1982). Subsequent advances in satellite remote sensing technology enabled a more detailed redefinition of the GL position, employing numerical terrain
models that incorporated SAR imagery from the ERS-1 and -2 missions complemented by data measuring ice thickness and density (Fricker et al., 2002). The evolution of satellite data use has markedly improved the precision with which we map the GLs on the AmIS. Techniques have evolved to include a combination of Synthetic Aperture Radar Interferometry (InSAR), Moderate Resolution Imaging Spectroradiometer optical imagery, and Ice, Cloud, and land Elevation Satellite (ICESat) laser altimetry data (Fricker et al., 2009). By employing data from multiple satellite missions, such as Landsat-8 and ICESat, studies
have investigated the stability of the AmIS's GL's dynamics, showing that its location is relatively stable over the observed time periods (Xie et al., 2016). Earlier studies of the AmIS iceberg calving cycle predicted that the ice shelf may not experience a major iceberg calving until around 2025 or later (Fricker et al., 2002), and on 25th September 2019 a calving event was observed producing the D-28 iceberg, which was the largest iceberg calving event on this ice shelf since 1960 (Francis et al., 2021). Collectively these results highlight the need for continuous satellite observations of Antarctic ice shelves and studies
that monitor major change.

## 2.2 Differential Range Offset Tracking

DROT is a dynamic method for GL detection that measures tidally induced vertical displacement of the ice shelf surface, by applying the differential principle of DDInSAR (Rignot, 1998) to fields of horizontal ice velocity obtained from SAR intensity offset tracking (Hogg, 2015; Joughin et al., 2016; Marsh et al., 2013; Wallis et al., 2024). To produce the DROT measurements
we used the following four steps:

   **i. SAR Image Acquisition:** The Sentinel-1 satellite constellation, a cornerstone of the EC-ESA Copernicus Programme, represents a leap forward in Earth observation capabilities (Snoeij et al., 2009). Comprised of two satellites, Sentinel-1a and Sentinel-1b (which ceased operating on 23rd December 2021), this system is equipped with an advanced C-band SAR instrument. The high-resolution, global coverage SAR images are acquired irrespective of weather or daylight conditions and
are used for a myriad of applications including maritime surveillance, land surface monitoring, and disaster management. Sentinel-1 satellites facilitate consistent monitoring of specific areas, including the polar region, with a repeat cycle of 12-days when operating independently or 6-days when the two satellites acquire data in tandem. In this study, we utilize the Sentinel-1 images acquired in the interferometric wide swath (IW) mode to measure the GZ. In IW mode, Sentinel-1 satellites employ a technique called Terrain Observation by Progressive Scans to steer the satellite's radar antenna to obtain multiple swaths of
data, which are then combined to form a single wide image. With a swath width of approximately 250 km, the IW mode offers a balance between swath coverage and spatial resolution, which is approximately 5 meters by 20 meters in range and azimuth directions respectively, for dual-polarization data. The dataset utilized in this study was collected over one-year period, beginning in January 2021 and concluding in December 2021.



**ii. Intensity Offset Tracking:** We employed the SAR intensity offset tracking method (Strozzi et al., 2002) to measure the
slant-range and azimuth registration offsets between a pair of SAR images. This method generates offset fields by employing
normalized cross-correlation on patches from the real-valued SAR intensity images (Paul et al., 2015; Strozzi et al., 2002).
The accuracy of detecting local image offsets is contingent upon the existence of nearly identical features within the SAR
images, such as crevasses and unique radar speckle patterns. When the images maintain coherence, the speckle pattern within
them can be correlated, allowing for highly accurate intensity tracking even with smaller image patches. Co-registration was
achieved using precise orbit data available 20 days post-image capture, yielding an azimuth co-registration accuracy of 0.001
pixels. This precise alignment allows for the orbital offset to be effectively discounted, isolating the movement attributable
solely to glacier dynamics. The geometry of a side-looking, off-nadir SAR sensor is such that a horizontal component is
measured if any vertical displacement occurs in the slant-range direction. The size of this displacement depends on the angle
from the local vertical at which the sensors signal intersects the Earth's surface, and the amplitude of any real vertical
displacement. Consequently, the slant-range displacement fields we obtain over areas of floating ice simultaneously capture a
real horizontal displacement caused for example by the flow of ice, and real vertical displacement originating from tidal motion
of the floating ice. For each single intensity range offset tracking result, when translated to ground range, the tidal bias S can
be estimated as the tidal change divided by the tangent of the incidence angle $S = \frac{\zeta_2 - \zeta_1}{tan\theta}$. $S$ is the tidal bias, $\zeta_2$ and $\zeta_1$ are the
true tide amplitude at the moments of SAR image acquisitions, $\theta$ is the incidence angle of SAR sensor, which is defined as the
angle between the radar beam and the local vertical at the point where the radar beam intersects the Earth's surface. For
example, at an incidence angle of 30º, a 1 metre tidal change between two SAR image acquisitions generates a 1.73 metre bias
in the horizontal displacement. In this study, we used the CATS2008 Antarctic tide model (Padman et al., 2008) to generate
ocean tide amplitude estimates. This ocean tide model estimates tidal movements within ice shelf cavities, achieving a
noteworthy precision of 5 cm (Glaude et al., 2020). To address the inverse barometer effect (IBE), the ocean's direct response
to variations in atmospheric pressure, we applied a conversion rate of 1 cm/hPa to our data (Padman et al., 2003). We derived
the IBE adjustments from anomalies in mean sea level pressure, ascertained through the fifth generation of ECMWF
atmospheric reanalyses of the global climate (ERA-5). The ERA-5 dataset offers hourly outputs at 31 km horizontal resolution,
from the Earth surface to 0.01 hPa (Hersbach et al., 2020). By computing the average atmospheric pressure throughout 2021
we established a baseline against which pressure anomalies could be compared, enabling us to determine the necessary IBE
corrections. These corrective measures are essential in refining our tidal models, providing a more accurate representation of
ocean surface height variations due to atmospheric pressure fluctuations, and thereby enhancing the precision of our analyses
(Chen et al., 2023).

**iii. Displacement Differencing:** A single Range Offset Tracking (ROT) displacement field over floating ice encompasses
both the actual horizontal displacement from ice flow and a signal caused by vertical tide motion. As displacement from ice
flow is much larger than tidal displacement, it would not be possible to measure the GL position from this data alone. In
methodology analogous to DDInSAR, we therefore difference two ROT displacement fields derived from the identical satellite



pass, to produce a DROT measurement where constant ice flow displacement is removed, and the ocean tide motion signal is isolated. If the two images of each pair share the same time interval, differencing the datasets will remove the consistent horizontal displacement (assuming a steady ice velocity), whereas the ocean tide amplitude which is not constant through time

will be visible as an anomaly (Joughin et al., 2016; Marsh et al., 2013). Depending on whether the two ROT displacement fields are constructed from three or four input SAR images, the differential tidal bias, $\Delta S$, within a DROT displacement field relies on the sea levels, $\zeta_n$, captured at the time of each SAR acquisitions:

$$\Delta S = \frac{(\zeta_3 - \zeta_2) - (\zeta_2 - \zeta_1)}{tan\theta} = \frac{\zeta_3 - 2\zeta_2 + \zeta_1}{tan\theta} \text{ (case of 3 input images)} \tag{1}$$

$$\Delta S = \frac{(\zeta_4 - \zeta_3) - (\zeta_2 - \zeta_1)}{tan\theta} = \frac{\zeta_4 - \zeta_3 - \zeta_2 + \zeta_1}{tan\theta} \text{ (case of 4 input images)} \tag{2}$$


**iv. GL Measurement:** To measure the GL location, the displacement fields from DROT must be distinctly divided into regions influenced by tidal bias and those that are not. In this study we do this by manually delineating the inland limit of tidal flexure in each DROT image, with the landward location where the displacement gradient begins to exceed zero determined to be the GL position. In the seaward direction, the point at which this gradient returns to zero is interpreted as point H (as

defined in Figure S1). Using both these positions we can obtain the width of the GZ from each DROT displacement field. Previous studies have used a threshold set close to zero to more automatically locate the position of GL (Wallis et al., 2024). However, in some cases this can be more complex as the ice shelf surface may exhibit considerable variations due to processes such as snow deposition, redistribution or surface melt, and surface deformation caused by ice flow. To ensure the GL is precisely located over a range of time periods we analysed the deformation gradient obtained from each DROT measurement.

**3 Results**

**3.1 Grounding Zone Distribution**

To detect the GLs, we construct DROT displacement fields using data acquired by the Sentinel-1a and -1b in 2021 following the approach introduced in section 2.2. The capability to produce multiple GL observations under varying tidal conditions within a single calendar year facilitates the delineation of the GZ and improves the accuracy of longer-term GL migration

measurements. We conducted the analysis using SAR images with a temporal baseline of 6-day, corresponding to frame 003 (orbits: 0834, 0839, 0843, 0848), with detailed information of selected SAR images presented in Table S1. We successfully produced a comprehensive GL dataset spanning 1,172 km along the majority of the AmIS coastline (Figure 1a and b). We extracted DROT deformation measurements along 18 profiles located across the GZ (Figure 2a to r), which clearly shows the vertical displacement of the floating ice shelf alongside the absence of any displacement on the inland section of each profile

for each period. This enables us to measure the inland limit of flexure experienced by the ice shelf and therefore delineate the GL location. We quantify the GZ width (as defined in Figure S1) by measuring the distance between the most seaward and



landward GL positions, tracing the trajectory of their migration. The width of the GZ derived from DROT, and its spatial variability around the AmIS coastline is also shown (Figure 1c), which unveils a zone of ephemeral grounding extending over 2,533 km$^2$ area. The limit of tidal flexure has migrated inland in 2021 by a range spanning several hundred meters to 14.2 km

(Figure 1c). Previous studies suggested that the GZ typically extends only a few hundred meters (Brunt et al., 2011; Christianson et al., 2016). However, with advances through data acquired by the ICESat-2 laser altimetry satellite, recent research has shown that this range can reach several kilometres. For example, tidal-induced GL migration of up to 15 km was observed at the Bungenstockrücken ice plain, far exceeding earlier estimates (Freer et al., 2023). This highlights the short-term variability of GZ dynamics, particularly in areas with significant tidal influences like AmIS. The GZ of AmIS varies in width

depending on the coastal geometry. It is narrower (< 4 km) in areas with a smooth, straight coastline, such as profiles 1 and 2 (Figure 1b and Figure 2a and b), and wider in regions with indentations or constricted geometry, as seen in profiles 5, 14, and 15 (Figure 1b and Figure 2e, n, and o). This suggests that the GZ expands in more confined areas. Nevertheless, the total short-term GL migration distance could potentially be still larger than we have captured in this study in regions where the full tide range has not yet been sampled by the SAR acquisitions used in this study.





**Figure 1.** (**a**) An example of a DROT result produced from Sentinel-1 data acquired on 08/03/2021, 14/03/2021 and 20/03/2021, with the colour scale corresponding to the amount of tidally induced vertical displacement. The location of zoom maps is also indicated (red box); (**b**) Chronological GL positions (coloured lines) mapped from 32 DROT results measured between January and November 2021. The colour gradient defines the acquisition date of each DROT measurement, with satellite data acquisition dates listed in Table S1. The location of cross-grounding zone profiles, labelled 1 to 18, is also shown (solid black line); (**c**) Summary map of GZ distribution (purple shading) for the AmIS, throughout the 11-month study period. Labels 'a' to 'k' indicate the locations of the 11 DROT-derived pinning points measured on the AmIS. The Reference Elevation Model of Antarctica (REMA) (Howat et al., 2019) is used as a basemap in subplots (**a**) to (**c**). (**d**)-(**f**) Zoomed in maps of the DROT displacement map for three locations labelled A to C in (**a**).





**Figure 2. (a-r)** DROT-derived displacements extracted along 18 profiles located perpendicular to the grounding zone, as shown in Figure 1b. The colour of each line corresponds to the DROT measurement date (yellow is Jan 2021, purple is Nov 2021) (colours consistent with scale used in Figure 1b), and light purple shading indicates the width of the measured GZ. **(s)** Distribution of tide amplitudes recorded through the year-long study period (light blue bars), shown alongside tide amplitude at times of each SAR image acquisition (grey dots).

Our results also show evidence of longer-term more permanent GL migration above the range of shorter-term tidal variability. In the region B marked by the red box in Figure 1a (profile 4, Figure 2d), there is evidence that the limit of tidal flexure has migrated inland in 2021 by a range spanning 5 to 10 km (Figure 1c).



## 3.2 Comparison with Other Grounding Line Measurements

To assess the performance of the DROT method for GL detection we performed a comparative analysis with contemporaneous
GL measurements made using the DDInSAR technique. We manually delineated the GL location in double differential
interferograms produced from Sentinel-1 data acquired on the same dates as the DROT measurements used in this study (Figure
3a). Weather processes at the ice sheet and shelf surface such as surface melt, snowfall and blowing snow affect the coherence
of the DDInSAR images, therefore, it is not always possible to make GL measurements around the entire coastline of the AmIS
using this technique. Nevertheless, these results enable us to directly compare the tidal deformation and GL location observed
in the same region and time-period, using independent but complementary methods. Along the western and eastern coastlines
of AmIS we observe a clear distribution of dense interferometric fringes in the double differential interferogram which
corresponds to the GZ, and in line with previous studies we delineate the inland limit of these fringes to be the GL location.
However, in the upstream part of AmIS where the Fisher, Mellor, and Lambert Glaciers converge, the rapid flow of ice causes
a large displacement of the ice surface resulting in phase decorrelation, which makes it difficult to distinguish the GL boundary
between in the DDInSAR fringe patterns. Where GL measurements were available from both techniques, we directly evaluated
the performance of the DROT results quantitatively through a comparison to GL positions derived from both independent
measurements (marked as b to d in Figure 3a). Our analysis suggests that on average the DROT-derived GL positions are
generally more seaward compared to those from DDInSAR, so this should be considered when assessing change from GL
measurements produced from independent techniques (Figure 3b.ii to d.ii). The DROT-derived GL's alignment with the
DDInSAR measurements is shown to be precise with the difference ranging from 0.35±0.14 km to 0.42±0.26 km (Table 1),
and coupled with the low standard deviation this underscores the reliability of the DROT method for accurately monitoring
the DDInSAR GL location. This demonstrates the complimentary nature of both tidally sensitive GL measurement techniques
and shows that the DROT method can be used alongside DDInSAR or at times when interferometric coherence does not exist.





**Figure 3.** Comparison of DROT-derived GL and contemporaneous DDInSAR-derived GL for three locations on AmIS with high coherence: **(a)** A Sentinel-1 double differential interferogram from 31/05/2021-06/06/2021-12/06/2021, with the contemporaneous DROT-derived GL (black line), and reference lines (green, red and purple lines) also shown. These reference lines correspond to the profiles analysed in **(b.ii-d.ii)** and the distribution histograms in **(b.iii-d.iii)**. **(b.i-d.i)** Zoomed in map of three locations labelled b to d in (a). **(b.ii -d.ii)** Difference between DROT-derived GL and DDInSAR-derived GL the along the reference profiles (shown in Figure 3a), where positive values indicate



that the DROT-derived GL positions are located seaward of the DDInSAR-derived GL measurement. **(b.iii-d.iii)** Histograms show the distribution of differences between the two GL measurements along the reference profiles (shown in Figure 3a).

**Table 1.** Mean absolute offset and standard deviation between the DROT-derived GL and contemporaneous DDInSAR-derived GL measured from 31/05/2021-06/06/2021-12/06/2021. Positive values indicate that the DROT-derived GL is located closer to the ocean compared to the DDInSAR-derived GL.

| Region | Mean absolute separation (km) | Standard deviation (km) |
|---|---|---|
| b | 0.42 | 0.26 |
| c | 0.35 | 0.14 |
| d | 0.35 | 0.15 |

We compared our DROT-derived GL results with two existing published GL datasets to understand the differences between these products. The Synthesized GL dataset (Depoorter et al., 2013a) is a continuous GL feature that fully encompasses the AmIS coastline, and was produced by combining multiple GL data sources including those produced from the DDInSAR technique, and ICESat and Landsat-7 satellite missions. By combining measurements from multiple techniques, some of which do not directly measure vertical displacement of the ice shelf by ocean tides, this GL product overcomes gaps in the spatial coverage which often limits the coverage of GL products from individual techniques. However, in order to achieve this coverage, the timestamp of the Synthesized GL data product is relatively large covering the period from 1996 to 2009. We compared our DROT-derived GL results to the MEaSUREs Antarctic GL, Version 2 (MAGv2) dataset (Rignot et al., 2016), which was produced by applying the DDInSAR technique to SAR images acquired between 1996 to 2000 acquired by ERS and RADARSAT SAR satellite missions. This is one of the most widely used GL datasets, however, similar to our contemporaneous DDInSAR GL measurements its coverage of the AmIS is incomplete as it is limited by poor coherence in some locations. To minimise the impact of tidal variations on GL detection, our comparison is based on the proximity of GL position to the landward and seaward boundaries (Figure S2), which enables a targeted evaluation of the DROT method's performance relative to established datasets under varying tidal conditions. The comparison of our DROT-derived GL with the MAGv2 shows that there is a 1.59±2.16 km mean absolute difference in the landward limit of the DROT-derived GZ, and 1.45±1.79 km difference for the seaward limit (Table 2). The comparison of our DROT-derived GL with the Synthesized GL shows a larger mean absolute difference of up to 1.55 km for landward limit of the GZ and 1.27 km for seaward limit of the GZ. These differences between the two published GL datasets and our DROT GL position are significant and are 311 % and 281 % larger (respectively) than the difference between the DROT and contemporaneously measured DDInSAR GLs (Table 1). The large size of this difference suggests that at least some of the disagreement is likely to be caused by real changes in GL over time, caused by the fact that all three products have different time stamps and are therefore not a direct like for like comparison. The cause of these real, time-driven GL location differences is likely to be linked to both short-term tidal variations





along with any longer-term migration caused by ice dynamic processes. Such short-term tidal variability can introduce

considerable change (up to 14.2 km) in the GL location as the position is highly sensitive to changes in sea level caused by tidal fluctuations. The MAGv2 and Synthesized datasets likely measure the GZ at different tidal states from the DROT GL observations, however, as precise timestamp information was not provided in these published products for every point along the AmIS coastline it's not possible to quantify the impact of this in more detail. Longer-term, more permanent migration of the GL caused by ice dynamic processes has been measured at a maximum rate of 1.3 km/yr on the most rapidly evolving

regions of West Antarctica (Park et al., 2013), however, this driver of change is thought to be substantially less pronounced in the AmIS region of East Antarctica. Nevertheless, this study has reported localised regions of GL retreat on AmIS, in region B shown in Figure 1(a), therefore we cannot rule out some contribution from this longer-term process in our comparison.

**Table 2.** Mean absolute separation and standard deviation between the DROT-derived GL and other comparable GL datasets

for the AmIS region, including the MAGv2 (Rignot et al., 2016), and the Synthesized GL (Depoorter et al., 2013b). The terms "landward GL" and "seaward GL" refer to the GLs closest to the ocean and closest to the land, respectively, among the 32 DROT-derived GLs. Positive values indicate that the DROT-derived GL is located closer to the ocean compared to the two published datasets.

| GL datasets | Landward GL | | Seaward GL | |
|---|---|---|---|---|
| | Mean absolute separation (km) | Standard deviation (km) | Mean absolute separation (km) | Standard deviation (km) |
| DROT vs MAGv2 | 1.59 | 2.16 | 1.45 | 1.79 |
| DROT vs Synthesized GL | 1.55 | 2.11 | 1.27 | 1.74 |
| MAGv2 vs Synthesized GL | 0.68 | 0.86 | 0.68 | 0.86 |

*Note: MAGv2 vs Synthesized GL values represent overall differences, not split by Landward/Seaward.


There is a level of uncertainty associated with the fact that the Synthesized GL product is produced from a variety of data sources and measurement techniques, including some that measure tidal displacement of the floating ice shelf and others that measure proxies for this such as the shadow caused by the break in surface slope at the GL. We attribute some of the difference between the DROT and the Synthesized GL datasets to these methodological differences. For example, the DDInSAR

technique identifies Point F as the proxy for GL whereas the surface slope methods (Hogg et al., 2018) designate Point $I_b$ as the GL location (Figure S1). While the DROT technique benefits from its ability to be applied to SAR amplitude images and therefore doesn't require interferometric coherence to be preserved, it has slightly lower measurement sensitivity (a fraction of a range pixel) compared to that of DDInSAR (half of a wavelength) (Joughin et al., 2016). Consequently, the DROT technique tends to position the GL slightly further seaward (1.5 km) than DDInSAR technique (Friedl et al., 2020; Joughin et

al., 2016), so this measurement bias must be accounted for when interpreting results from the different techniques. To investigate this point further we also compared the absolute difference between the MAGv2 and Synthesized GL products, to





quantify the difference between these two published products. This showed that there was 0.68±0.86 km difference, highlighting the value of having contemporaneously acquired data for GL validation and error characterisation exercises.

### 3.3 Modes of Tidal Grounding Line Migration

To probe the interplay between tide amplitude as estimated by tidal models and GL migration distance in our DROT results, we adopt the categorization framework established by Freer et. al (Freer et al., 2023), delineating three distinct tidal GL migration behaviours. We plot the results of the GL positions along 18 profiles versus the maximum tide amplitude (Figure S2) with highlighted profiles displayed in Figure 4a to i. Based on our visual assessment of the predominant patterns we categorize the migration modes of 18 profiles across different regions as asymmetric, threshold, and linear.

The most common mode observed in this work is linear, where 8 out of 18 profiles display a direct proportional relationship between maximum tide amplitudes and GL migration distances. For example, in profile 5, for every 1-m increase in maximum tide amplitude the GL migrates approximately 3.1 km upstream. The second mode is threshold which we observe in 3 out of the 18 profiles. This mode manifests as the GL remaining relatively stationary unless a specific tidal threshold is surpassed. For instance, the GL positions along profile 2 (Figure 4d) remain largely unchanged when maximum tide amplitudes are below 0.95 m but exhibits substantial upstream migration when the tide exceeds this threshold. Our results suggest that there is spatial variability in the value of the threshold; for example, at profiles 16 and 18 which are located on the western side of AmIS rather than the east, the threshold is lower at 0.62 m. Variation in this threshold is likely to be linked to factors including ice thickness, tide amplitude and bed geometry, however, further work is required to understand this in more detail now that the mode of variability has been characterised. The third mode of GL variability is asymmetric (3 out of 18), which is exemplified by profile 11 (Figure 4h) where the GL retreat rate correlates linearly with maximum tide amplitude, with a rate of 0.49 km per 1 m of tidal change when tides are below 0.62 m. As the maximal tide amplitude increases beyond 0.62 m the GL location jumps by 0.66 km and the subsequent GL migration increases to a rate of 1.24 km/m. Similarly, for profiles 7 (Figure 4g) and 17 (Figure 4i), when the maximum tide amplitude is less than 0.62 m the GL positions cluster together, indicating a lower rate of migration. When the maximum tide amplitude exceeds 0.6 m, there is a jump in migration and subsequent GL migration rates reach 1.42 km/m and 4.32 km/m respectively. This aligns with findings from previous studies (Tsai and Gudmundsson, 2015) which suggest that if we assume a constant surface and bed slope, the GL migration upstream during high tides is tenfold that of the downstream migration during low tides. Although our results do not show an upstream migration rate ten times greater than the downstream rate, larger upstream migration is observed which provides supporting evidence that high tides dominate the GL's ephemeral migration within a tidal cycle.





**Figure 4.** Tidal GL migration modes of AmIS. The top three rows depict the relationship between GL migration distance and maximum tide amplitude, which corresponds to the highest tide amplitude among three dates used to derive the DROT results, across linear **(a-c)**, threshold **(d-f)**, and asymmetric **(g-i)** migration modes. The bottom row **(j-l)** shows the relationship between GL migration distance and absolute differential tide range, defined as the absolute value of the difference between the tidal differences from two ROT results. The solid lines represent the results of linear regression applied to the data, and the dashed lines in the threshold mode indicate the critical values, beyond which significant shifts in GL migration are observed.



While most GL migration shows a correlation with tide amplitude, as illustrated by Figure 4a to i, the migration patterns of
the remaining four profiles (1, 10, 14, and 15) do not follow this trend (Figure S3). In these cases, GL migration does not
exhibit a clear relationship with maximum tide amplitudes, and sometimes even demonstrates greater upstream migration
during lower tides. This discrepancy could be explained by some studies which suggest that a linear relationship between GL
position changes and tide amplitudes only holds when tide amplitudes approach infinity (Tsai and Gudmundsson, 2015). In
the AmIS, the tide range is approximately between -1.2 m and 1.6 m, but the range captured from SAR images in 2021 spans
from -0.4 m to 1.5 m, covering only 64% of the full tidal cycle. Therefore, when tide amplitudes are not sufficiently extreme,
other factors, such as basal properties, surface slope, and ice thickness, may play a more significant role in affecting the
ephemeral migration of the GL. Furthermore, similar to the DDInSAR technique, DROT captures a tide range rather than a
specific tide amplitude. Using the maximum tide amplitude from the selected SAR images for discussing the relationship
between the tide range and ephemeral GL migration may not provide a comprehensive insight, therefore, we investigated the
relationship between double-differential tide range and the GL positions (Figure S3 and Figure 4j to l). We find that there is a
positive linear relationship between the temporal migration distances of the GL and the absolute double-differential tide range,
meaning that greater ice surface deformation corresponds to a more inland position. So far, there is no definitive conclusion
as to whether the maximum tide amplitude or the double-differential tide range predominately influences the GL migration
when it comes to double-differential methods (DROT and DDInSAR). The results of our study indicate that along these profiles
there are different migration modes in relation to the maximum tide amplitude, whereas a linear trend is observed when
considering the absolute double-differential tide range.

### 3.4 Regional Case Studies

### 3.4.1 Case A: The widest grounding zone observed in Amery Ice Shelf

Spatial analysis of the GZs on the AmIS reveals significant variation in the GZ width, with the widest section reaching 14.2
km upstream of its most seaward limit (box A in Figure 1a). Unlike the linear, asymmetric, and threshold modes of tidal GL
migration previously outlined, profiles 14 and 15 (Figure S3n and o) do not follow these patterns, with no clear link between
absolute GL positions and the highest tide amplitude observed over the study period. This finding challenges our hypothesis
that the most retreated GL positions correlate with the highest tides. Both profiles 14 and 15 lack a migration pattern that
correlates with the ocean tides, which aligns with the findings of Chen et al. (2023) who report a recurring pattern where the
GL alternates between two positions. After the GL stabilizes upstream for several weeks, it then transitions downstream. These
re-advances do not correlate with tide cycles, and the transitions do not coincide with periods of lower tides. However, the
simultaneous occurrence of these shifts in both profiles 14 and 15 suggests a more complex interplay with tidal states than
direct causation. We studied profiles 14 and 15 during different tide states over time, which revealed a consistent downstream
migration of 7 km over the initial three months of our study period (31/01/2021 to 25/04/2021), a nearly six-month stabilization
near this new position, followed by a swift retreat to the starting location within a month (Figure 5b). The absence of short-



term cyclic variations in this pattern presents a stark contrast to previous studies (Chen et al., 2023), indicating that factors other than tides are influencing GL migration in this region.

Considering that three major glaciers that converge in area A, the dynamic changes of these glaciers will directly affect the
grounding zone in this region (Figure 5a). A chain of active -lakes was discovered beneath Lambert Glacier with isolated subglacial lakes are thought to exist in the upstream regions of the Mellor and Fisher Glaciers and areas where the ice flow is relatively slow (Hogg et al., 2021; Wearing et al., 2024). While subglacial hydrological pathways are known to exist at the ice-bed interface, active subglacial lakes intermittently fill and drain, substantially altering the subglacial water pressure. When active subglacial lakes drain, they release large quantities of water, increasing the water pressure at the ice-bed interface. This
elevated pressure reduces friction between the ice and its bedrock, effectively lubricating the ice base, which can increase the velocity of ice flow towards the ocean. When cold, fresh subglacial water enters the ocean at the grounding line, it can cause plume driven turbulent melt at the ice shelf base, thinning the ice shelf in the vicinity of the GL or producing smaller scale channelized melt features (Gwyther et al., 2023; Nakayama et al., 2021). Drainage of large volumes of water from active subglacial lakes may lead to the formation of efficient subglacial channels that transport meltwater from the interior of the
continent to the GLs. These channels can deliver relatively warm water (warmer compared to the surrounding ice) directly to the ice shelf base (Goldberg et al., 2023), which can enhance basal melting by increasing the heat flux at the base of the ice shelf. Enhanced basal melting can thin the ice shelf and potentially destabilize it over time. The previous study (Adusumilli et al., 2020) elaborates on spatially averaged melt rates for AmIS, where rates varied from near zero to 6.5 m/yr, with maximum values observed between 2003 and 2007. This variability in melt rates is speculated to be linked to the drainage of a ~0.8 km$^3$
subglacial lake under Lambert Glacier, which can drive energetic plumes that increase basal melting rates near GLs. These observations highlight the importance of considering subglacial hydrological processes and their potential to influence GL stability beyond direct tidal interactions. The recurring, non-cyclic GL migration observed in profiles 14 and 15 could be partly explained by subglacial water fluxes that impact the GL position independently of tidal forcing, further emphasizing the complex drivers of GL dynamics in AmIS.








**Figure 5. (a)** Map showing basal melt rates (Davison et al., 2023) and ice velocity (Rignot et al., 2017) across AmIS, with zoomed-in views of three key regions (A, B and C) highlighted in the main map with red boxes. The coloured lines represent DROT-derived GLs at 32 dates, while the white line indicates the GL from MAGv2 dataset (Rignot et al., 2016). The light blue arrows represent the likely clockwise direction of ocean water circulation within the AmIS cavity. **(b)** Time series plot showing GL migration distance along two profiles (shown in zoomed-in maps of region A), where circles represent profile 14 and triangles represent profile 15. The colours correspond to those used in **(a)**. **(c)** Elevation along profile 4 (shown in zoomed-in maps of region B), with the red dashed line indicating the GL from MAGv2 and the purple shading between the two dashed lines representing the DROT-derived GZ in this study. **(d)** Elevation along the transect X-X' (shown in zoomed-in maps of region C), with the blue dashed line outlining the region between Gillock Island and the ice sheet, showing the sector's vertical structure. All elevation data are sourced from the MEaSUREs BedMachine Antarctica Version 3 (Morlighem, 2022).

The absence of permanent GL retreat at area A, despite its relatively high basal melt, can be attributed to several factors. The

basal melt of area A is driven by cold, dense high salinity shelf water and typically involves refreezing of some of the meltwater

back onto the ice shelf, which can mitigate direct contributions to sea level rise by maintaining a sort of balance at the GL

(Adusumilli et al., 2020; Jacobs et al., 1992; Motyka et al., 2013; Smith et al., 2009; Washam et al., 2019). In the case of the





AmIS, about a fifth of the meltwater generated from basal melting is refrozen as marine ice, which contributes to maintaining
the stability of the GL. It is precisely because the ice near the GZ undergoes a cyclical process of melting, freezing, and
remelting that the GL position may continuously change. The greater the ice shelf basal melt rate the more significant the
impact on this area is likely to be, and consequently, the larger the extent of the GZ becomes.

### 3.4.2 Case B: Long-term Grounding Line Retreat

Long-term GL migration is crucial for predicting the stability of ice sheets and their impact on global sea level rise. It serves
as a key indicator of the interactions between climate change and polar ice dynamics. Research on the AmIS since 2000 has
concentrated on understanding its mass balance, surface meltwater (Tuckett et al., 2021), ocean circulation beneath the ice
shelf (Wu et al., 2021), thermal structure and stability (Wang et al., 2022), and the effects of iceberg calving and basal melting
(Walker et al., 2021). However, studies focusing specifically on the dynamics of the GL of the AmIS are relatively scarce.
This indicates a potential gap in the literature where more focused studies could enhance our understanding of GL behaviours
in relation to broader climatic and oceanic influences. Here, we bridge this gap by comparing our DROT-derived GL which is
based on Sentinel-1 images from 2021, to the composite MAGv2, which, in the AmIS region was derived from RADARSAT
images acquired in 2000. Our comparative analysis reveals a significant retreat of the GL in area B (Figure 5c), with the
maximum retreat reaching up to 5 km compared to the seaward DROT-derived GL and 10 km compared to the landward
DROT-derived GL. This observed GL retreat in area B may be significantly influenced by high basal melt rates. The AmIS
experiences elevated basal melt rates in specific regions, particularly along profile 4 (Figure 5a), where average melt rates are
reported at 8.3 m/yr (Davison et al., 2023). This high melt rate likely contributes to GL retreat by thinning the ice shelf, which
reduces buttressing and promotes imbalance near the GZ. Several oceanic models have confirmed a clockwise circulation
within the AmIS ice cavity (Herraiz-Borreguero et al., 2015; Liu et al., 2018), moving from east to west (the light blue arrows
in Figure 5a) through Prydz Gyre. The basal melting is further influenced by the inflow of warm modified Circumpolar Deep
Water on the eastern side of the AmIS, contributing to increased basal melting on the eastern AmIS, reducing ice shelf
thickness, and thus promoting GL retreat.

In addition to basal melting, the observed GL retreat in this region may be influenced by the drainage of supraglacial lakes.
Widespread supraglacial lake networks have been documented up to 30 km upstream of the GZ in this region, and satellite
observations show that they have reoccurred up to 14 times between 2005 and 2020, suggesting consistent seasonal surface
melt (Tuckett et al., 2021). On grounded ice, lakes commonly develop in areas with localized melt enhancement and relatively
low accumulation rates, often located near rock outcrops and blue ice. The regular formation of surface lakes in the same
location each year over decades indicates that surface melt is a persistent factor in the region's glaciological environment, with
potential implications for ice flow and other ice dynamic parameters such as the GL (Kingslake et al., 2017; Langley et al.,
2016). When supraglacial lakes drain, potentially through processes such as hydrofracturing, meltwater released may lubricate
the glacier bed, accelerating ice flow and possibly impacting the GL location.





The retreat of GLs is also significantly influenced by whether the underlying bed slope is retrograde or prograde (Joughin et al., 2014; Schmidt et al., 2023; Schoof, 2007). Along the GL retreating path, the underlying bed slope of the first half is in a
retrograde state (Figure 5c). As the ice retreats into deeper water, more of the ice shelf lifts off the bed and becomes afloat, reducing frictional resistance and promoting further retreat. However, in the latter half, to the south, it slopes upward as it extends inland from the GL, which generally offers more stability, slowing down GL retreat and providing a buffer against rapid changes.

### 3.4.3 Case C: Intermittent Grounding

Our observations have identified an anomalous peninsula of the AmIS coastline situated between Gillock Island and the ice sheet, where we observe relatively short-term transitions between grounded and ungrounded states. Published GL datasets, like the MAGv2 and Synthesized GL, have consistently shown this area as grounded; however, our DROT-derived data from 2021 definitively shows that the region is both grounded and ungrounded at different time periods. For example, our results show that between 31/01/2021 and 24/04/2021 the region is ungrounded, between 19/05/2021 and 06/06/2021 the region is
grounded, and then between 12/06/2021 and 30/06/2021 the region is ungrounded again. This cyclical behaviour is atypical for Antarctic coastline regions and might be influenced by the subglacial topography and oceanographic dynamics. Bed topography data (Morlighem, 2022) show that a 447 m deep trough is present at the 'neck' of the peninsula (Figure 5d), which could channel warmer ocean water that may enhance basal melt rates. This process intermittently reduces the ice's grounding, leading to periods where Gillock Island acts as an isolated pinning point. While basal melt variability provides one plausible
explanation, it is important to consider ocean tides as an additional factor influencing grounding. Tidal fluctuations may cause periodic changes in water pressure beneath the ice shelf, which can alternately ground and unground regions close to the threshold of flotation (Padman et al., 2018; Rignot et al., 2011). During high tides, buoyant forces could lift the ice, causing ungrounding, while low tides might allow the ice to settle back onto the bedrock. This tidal influence, especially near regions with topographic lows (Figure 5d), can significantly impact GL dynamics and might work in tandem with basal melting to
drive the observed oscillatory grounding behaviour (Gudmundsson, 2011; Robel et al., 2017). Such observations underscore the complexity of ice sheet dynamics and emphasize the influence of subglacial and oceanographic features in determining the stability and response of ice structures in polar environments. Incorrect determination of the GL is important as it could lead to inaccuracies in models predictions of ice dynamic behavior, which are essential for forecasting sea level rise (Andrew et al., 2018; Rignot et al., 2014).





## 4 Discussion

### 4.1 Advantages and Limitations of the DROT Method for Monitoring Grounding Line Migration

The prevalent remote sensing methods for investigating the temporal migration of GL positions primarily include RTLA which is based on ICESat and ICESat-2 datasets, and DDInSAR, which utilizes SAR imagery. In comparison to RTLA, the DROT method used in this work has the advantage that it has i) all-weather measurement capability as SAR images penetrate through cloud cover and precipitation, enabling effective operation under poor weather conditions that would typically impede laser altimetry; ii) higher spatial resolution although RTLA has a finer along-track resolution of 20 m (Freer et al., 2023) compared to DROT's 100 m (Wallis et al., 2024), DROT offers a more continuous spatial measurement, as RTLA has data gaps every 2.8 km due to the track spacing (Abdalati et al., 2010). This enables DROT to capture fine-scale GL features across larger areas without interruption. Finally, iii) DROT GL measurements can be made at fine temporal resolution every 6 to 12 days vs the typical 91 days repeat period for laser altimetry missions, providing more frequent and continuous temporal coverage which is essential for monitoring short-term tidally induced GL migration. The DDInSAR technique shares many of the advantages of DROT outlined above, however, its requirement for high coherence which can be easily lost, limits the extent to which this method can be used. Coherence is reduced if the scattering properties of the measured surface are not stable between two subsequent acquisitions, which frequently takes place due to snow deposition, snow redistribution or surface melt, as well as due to fast ice flow. The main benefit of DROT over DDInSAR is the ability to apply this technique in low-coherence areas.

A limitation of the DROT method is that unlike RTLA the DROT method cannot detect the GL position at a specific time point but instead measures the location within the tide range that occurs between two image acquisition dates. Prior to the launch of Sentinel-1 in late 2014 SAR images were not routinely acquired over the whole Antarctic coastline limiting the ability to apply techniques such as DDInSAR or DROT that require these input datasets. In contrast, altimetry missions such as ICESat-2's have an orbit covers the entire Antarctic coastline yielding a more comprehensive spatial coverage, even if the spatial resolution is coarser. As new SAR missions such as the NASA-ISRO NISAR and ESA-Copernicus ROSE-L missions are launched, this will add to the volume of SAR data already acquired by Sentinel-1, ensuring excellent spatial and temporal coverage. All tidally sensitive GL measurement techniques perform best in regions with a high tide amplitude, where the boundary between floating and grounded ice is more easily distinguished above any measurement noise. The DDInSAR and RTLA techniques have more sensitivity to vertical displacement of the ice surface than the DROT technique, however, there is also a range of performance for the DROT method when applied to different radar frequency SAR data, with shorter wavelengths (e.g. X-band SAR) having more sensitivity to the altitude of ambiguity than longer wavelength (e.g. C-band SAR). The most significant limitation of DROT is the inability to distinguish between actual tidal and non-tidal-driven velocity variations, and the artificial tidal bias. Non-tidal-driven velocity variations refer to changes in ice flow speed that are not caused by tidal forces, such as seasonal changes in ice dynamics, subglacial water drainage, or adjustments in ice shelf stress. This



artificial tidal bias arises from vertical displacement of the ice shelf due to tidal lifting and lowering, which, when measured by satellite radar, appear as horizontal movements along the radar's ling of sight. Since DROT can only track movement in the direction of the radar signal, these vertical shifts are misinterpreted as horizontal migration of the GL, even though they are not true horizontal movements caused by tidal forces. This projection effect can lead to errors or distortions in the results, potentially causing misinterpretations about the true position and movement of GL (Friedl et al., 2020).

## 4.2 Potential for Measuring Pinning Points

A significant amount of ice loss from the Antarctic Ice Sheet can be attributed to the effects of warm ocean currents undermining the buttressing capability of ice shelves, leading to their thinning and increased basal melt (Jenkins et al., 2010, 2018; Pritchard et al., 2012). This process has direct implications for the acceleration of ice discharge into the ocean (Greene et al., 2022; Gudmundsson et al., 2019). Therefore, comprehensive observations that monitor the changes in ice-shelf properties are crucial for understanding the evolution of the Antarctic Ice Sheet and for predicting future patterns of ice loss. Changes in the surface expression of pinning points, as a proxy for ice-shelf thickness change, could assist in reducing uncertainties regarding Antarctica's impact on global sea level (Edwards et al., 2019; Miles and Bingham, 2024; Ritz et al., 2015). Pinning points are locally grounded features on ice shelves that provide additional buttressing to the ice flow (Matsuoka et al., 2015). They are regions where the ice shelf is anchored to the bedrock or underlying topography, providing structural stability to the shelf and resisting the flow of ice (Jenkins et al., 2010; Roberts et al., 2018). Larger pinning points provide more substantial buttressing against the flow and bending of ice shelves, thereby slowing down the ice flow and enhancing stability. Their presence can influence the pattern of ocean circulation beneath the ice shelf, which, in turn, affects the basal melting patterns. Moreover, large pinning points can cause the ice flow to diverge around them, creating complex flow structures within the ice shelf. Consequently, understanding the role of pinning points and measuring any change in their area helps to assess the stability of ice shelves and the potential for rapid ice-sheet changes.

Figure 1c clearly showcases the potential of the DROT technique for measuring ice shelf pinning points in the AmIS, including their location, area, and grounding conditions during various tidal states. Our findings show that DROT results can be used to identify 11 pinning points on the AmIS (labelled a-k according to the direction of ice flow in Figure 1c), including 6 new features which are not included in the MAGv2 dataset. The occurrence of pinning point g is unstable; as discussed in section 4.1, the intermittent grounding between Gillock Island and the ice shelf causes it to switch between connected and disconnected states with the ice sheet. The absence of pinning points h to k in the DDInSAR GL data product is unknown, but may be due to their location near the ice shelf edge which may have made them more difficult to identify in the interferograms. Another possibility is that the coverage of historical SAR datasets in this region was poor, leading to a more limited number of coherent InSAR, making it less likely that the pinning points were identified. Compared with other updated datasets of pinning points, some of these pinning points have been identified (Figure S7). For example, we note that in the more recent DDInSAR GL dataset (Mohajerani et al., 2021) these pining points (f, g, h, and j) were identified, but pinning points (e, i, and k) were not.



The latest optical imagery-based pinning points dataset (Miles and Bingham, 2024) successfully captured pinning points (h, i, j, and k), but missed pinning points (a, c, f, and g). This suggests that when more data and automated techniques are combined, lesser studied features are identified. In addition to mapping the number of pinning points, our DROT results allow us to measure the pinning point area over time and at different tidal states. For example, the area of pinning point 'a' fluctuates
545    between 170 and 289 km$^2$ in the 32 DROT results we sampled. Summing up all 11 pinning points within the AmIS, their combined minimum area totals 639 km$^2$, while the maximum combined area reaches 956 km$^2$. The size of pinning points has a direct impact on their buttressing influence therefore understanding how pinning point area varies over time will be important for modelling studies. In summary, the DROT results effectively identify the locations and areas of ice shelf pinning points, and there is an opportunity for future studies to use this technique to monitor pinning points on ice shelves across Antarctica.

550    **4.3 Factors Affecting the Grounding Zone Width**

As illustrated in Figure 1c, the GZ of AmIS exhibits varying widths depending on the coastal geometry. In areas with a smooth, straight coastline, like profile 1 and 2 (Figure 1b and Figure 2a-b), the GZ is narrower, typically less than 4 km. Conversely, in regions with indentations or more constricted coastal features, such as profiles 4, 14, and 15 (Figure 1b and Figure 2d, n, and o), the GZ becomes wider.  We note that these confined regions tend to cluster in areas of ice stream convergence or where
555    the ice velocity is higher. Therefore, we compare the GZ width along the profile direction with the average ice flow velocity (Figure 6a), and the results indicate a linear relationship; the greater the ice flow speed, the larger the GZ width. The specific geometry of an ice shelf and its interconnection with the inland ice also significantly impacts its tidal response. Areas where the ice shelf curves towards the ice sheet have a different stress distribution, leading to a more varied tidal response compared to more linear sections (Brunt et al., 2010). Faster ice flow suggests higher dynamism in that region, potentially leading to a
560    more sensitive response to external forces like tidal motion. Areas with rapid ice flow may experience greater stress concentration, making the ice shelf more prone to deformation under tidal forces. There can be complex interactions between the dynamics of ice flow and tidal forces, leading to more pronounced tidal responses in areas of faster ice flow (Brunt et al., 2010).



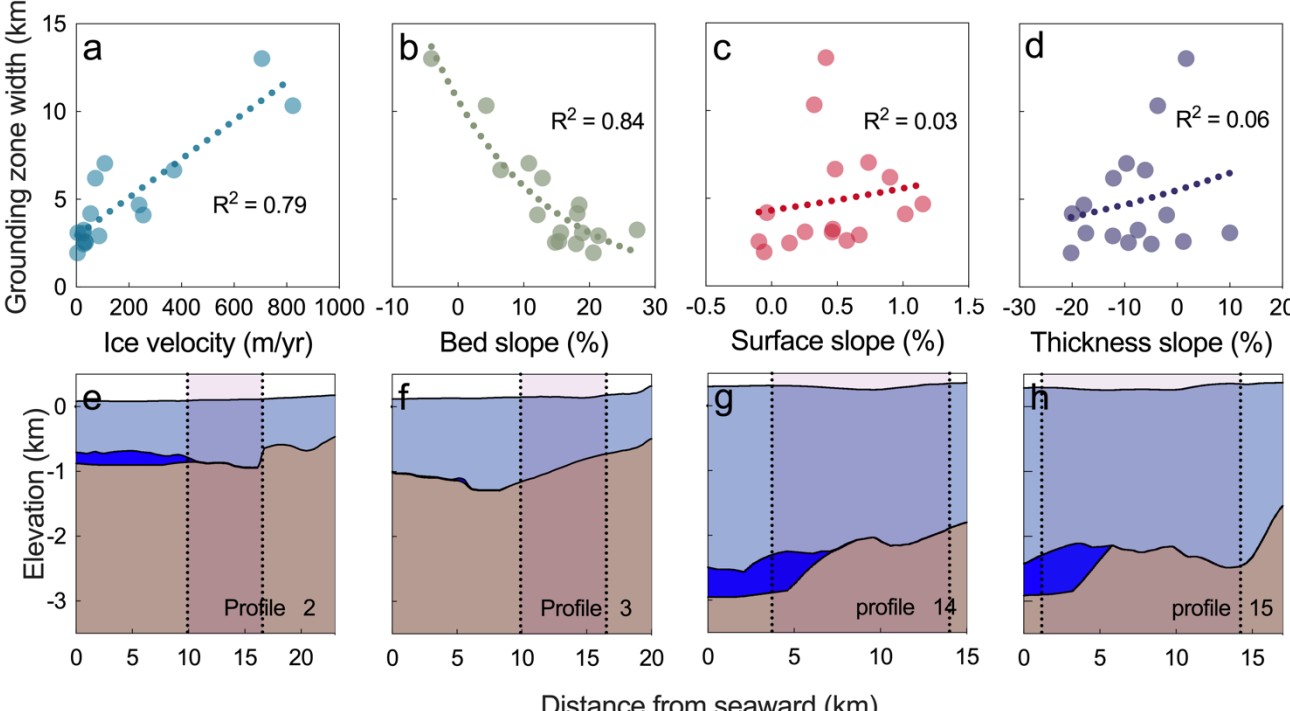

**Figure 6.** The top row **(a-d)** shows the correlation between GZ width and various glaciological parameters: **(a)** ice velocity (sourced from the MEaSUREs InSAR-Based Antarctica Ice Velocity Map, Version2 (Rignot et al., 2017)), **(b)** bed slope, **(c)** surface slope, and **(d)** ice thickness slope, all derived from the MEaSUREs BedMachine Antarctica Version 3 (Morlighem, 2022). The R² values indicate the strength of these correlations. The bottom row **(e-h)** presents the cross-sectional elevation profiles corresponding to profiles 2, 3, 14, and 15, with data soured from the MEaSUREs BedMachine Antarctica Version 3. The GZs marked by the shaded areas between the dashed lines.

Our study involves a comprehensive evaluation of the relationship between GZ width and the corresponding slopes of the bedrock (Figure 6b), surface (Figure 6c), and ice thickness (Figure 6d). In areas with a wide GZ, we observe that the bed slope typically exhibits either a retrograde or a mild prograde inclination. This is especially evident in profiles 14 (Figure 6g) and 15 (Figure 6h), where the seawater, upon traversing the initially gentle prograde slopes of the bedrock, progresses into an extensive area of retrograde slopes measuring several km in length, with slopes of 4.1% and 1.4% respectively. Conversely, for the majority of the profiles studied the seawater's forward motion is hindered by more pronounced prograde slopes at the onset, which exhibit steep inclines of between 21% to 61%. This suggests that the geometry of the initial bedrock incline plays a pivotal role in influencing the extent of seawater penetration into the GZ. To quantify the slopes associated with the GZ width, we have adopted a method of averaging the slopes located seaward and landward of the GL. The result indicates a quasi-exponential relationship between the GZ width and the bed slopes (Figure 6b), indicating that as the bed slope becomes gentler, the GZ width tends to expand significantly. Notably, however, this correlation does not extend to the surface slope or ice thickness slope. Our analysis does not reveal any obvious mathematical relationship between these parameters and GZ width, suggesting that other factors or complex interactions may govern the influence of surface and ice thickness slopes on



GZ width. This absence of a clear link highlights the need for further research to understand the interplay of these variables and the dynamics of GZ stability and migration.

## 5 Conclusion

Our study has used the DROT technique to measure the short-term migration of the GL under the influence of tidal variations on AmIS. We used the DROT method to observe GL positions at different tidal states throughout 2021 producing a comprehensive map of the GZ distribution across the ice shelf, which fluctuated significantly in response to tides ranging from -0.5 metres to 1.5 metres. Our results perform well compared to the contemporaneous DDInSAR measurements, with a mean absolute separation between these data of 0.37 km and a standard deviation of 0.19 km. We analyse our results to show three different modes of GL migration, (linear, asymmetric, and threshold), in relation to the maximum tide amplitude, and we show there is a positive linear trend when considering the absolute double-difference tide range. Our results show that beyond ocean tide driven variability, factors such as basal melting, subglacial topography, and ice flow velocity play significant roles in influencing GL migration patterns. These findings reveal localized variations in GL behavior, including evidence of long-term GL retreat of 5 to 10 km in area B (Figure 5) of AmIS, highlighting the complex interplay of environmental and glaciological factors on GL stability and migration. In addition, we found that the DROT technique can be used to accurately identify the location and area of ice shelf pinning points, which could be used a proxy for ice-shelf thickness change. Looking forward, studies should extend the application of the DROT technique across Antarctica, and combine these results with other remote sensing techniques to enrich the dataset available for studying the Antarctic grounding zone. Only through a multifaceted approach can we hope to gain a holistic understanding of the processes governing the GZ and, by extension, the health of the entire Antarctic Ice Shelf system.

*Code availability.* The authors have made available, for the proposes of review, code used in DROT processing in this article is at https://zenodo.org/doi/10.5281/zenodo.14912749.

*Data availability.* The 32 DROT-derived GLs of AmIS and double-differential interferograms used for evaluation have be made available at https://zenodo.org/doi/10.5281/zenodo.14912749.

*Author contributions.* Y.Z., A.E.H. and A.H. conceived the study, Y.Z. performed the analysis and wrote the paper. All authors discussed the results and contributed to the preparation of the paper.

*Competing Interests.* The contact author has declared that none of the authors has any competing interests.



*Acknowledgements.* The authors gratefully acknowledge the European Space Agency (ESA) and the European Commission
for the acquisition and availability of Copernicus Sentinel-1 data. Data processing was undertaken on ARC3 and ARC4, which
are part of the high performance computing facilities at the University of Leeds, UK. Funding is provided to A. E. Hogg by
ESA through the Polar+ Ice Shelves project (grant no. ESA-IPL-POE-EF-cb-LE-2019-834) and the SO-ICE project (grant no.
ESA AO/1-10461/20/I-NB), by NERC via the UK EO Climate Information Service (grant no. NE/X019071/1). We
acknowledge the support through a China Scholarship Council awarded to Y. Zhu (202206270074). B. J. Wallis was supported
by the Panorama NERC Doctoral Training Partnership (DTP), under grant NE/S007458/1 and by NERC through the UK EO
Climate Information Service NE/X019071/1.

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
