# Peer review of "Short and Long-term Grounding Zone Dynamics of Amery Ice Shelf, East Antarctica"

_EGUsphere, 2025_

## Author Comment (AC1)

**Response to reviewer #1 for 'Short and long-term grounding zone dynamics of Amery Ice Shelf, East Antarctica' - EGUSPHERE-2025-849**

Yikai Zhu, on behalf of the authors,

We would like to thank Reviewer #1 for their detailed and constructive feedback on our manuscript. We greatly appreciate the time and effort taken to read the manuscript and provide insightful suggestions. These comments have been very helpful in improving the quality and clarity of our work. Below, we provide a point-by-point response to each of the reviewer's comments. Reviewer comments are reproduced in the **Comment** column, with our **Response** listed alongside. The **Line** column refers to the position of the relevant text in the original submission, and the **New line** column indicates where changes were made in the revised version. All modified text is highlighted in blue to clearly indicate revisions made in response to reviewer feedback.

| ID | Comment | Line | Response | New line |
|---|---|---|---|---|
| 1.1 | In particular, the identification of linear, threshold, and asymmetric migration modes largely replicates the classification framework introduced by Freer et al. (2023), and the reported correlations between GZ width and glaciological parameters (e.g., bed slope, ice velocity) parallel earlier insights from Chen et al. (2023). | | We appreciate the comment and gratefully acknowledge the classification framework introduced by Freer et al. (2023), which provided a useful reference for analyzing short-term grounding line migration patterns. Similarly, the insights from Chen et al. (2023) were valuable when examining potential controlling factors. The inclusion of ice velocity as a parameter in our analysis represents a new addition in this study. | |
| 1.2 | This work provides a high-quality observational foundation and demonstrates the utility of DROT for grounding line science. | | We sincerely thank the reviewer for the positive comment. | |
| 1.3 | Consider elaborating on the specific advantages of the DROT method in this context and providing a brief rationale for its selection over other grounding line detection techniques. This would help clarify the methodological motivation at an early stage in the manuscript. | | **Done.** In response, we have revised the introduction to briefly explain the rationale for selecting the DROT method in this study. Specifically, we now highlight that DROT enables high-frequency, spatially continuous measurements under all-weather conditions, and offers advantages over other techniques: it is less affected by cloud cover and track spacing limitations compared to RTLA, and remains effective in fast-flowing and low-coherence areas where DDInSAR technique is often limited. We believe this revision helps clarify the methodological motivation at an early stage in the manuscript.

Line 41: "In this study, we adopt the DROT method due to its ability to provide spatially continuous, high-frequency measurements under all weather conditions. Compared to RTLA, it is less affected by cloud cover and track spacing limitations, while unlike DDInSAR, it remains effective in fast-flowing or decorrelated regions where interferometric coherence is often lost." | 41-44 |

| | | | | |
|---|---|---|---|---|
| 1.4 | It would be helpful to clarify the basis on which the GL migration along profile 4 is interpreted as permanent. Specifically, how does Figure 2d/Figure 1c support the conclusion that the observed inland migration exceeds short-term tidal variability? | 220-222 | **Done.** We have revised the text in this section to clarify our criteria for identifying long-term GL migration. Specifically, we compared the historical GL location from MAGv2 dataset with the seaward limit (Fmax) of the DORT-derived GL positions in 2021. If the historical GL is located more than 4 km farther seaward than our most seaward DROT observation, we interpret this as an indication of long-term retreat beyond short-term tidal variability. In profile 4, the MAGv2 GL is located over 5 km farther seaward than the DROT-derived Fmax, which supports our interpretation of a sustained inland migration in this region.

Line 221: "To determine whether this inland migration represents a long-term change beyond short-term tidal variability, we compare our DROT-derived seaward GL position (Fmax) with the historical GL location from the MAGv2 dataset. If the reference GL lies more than 4 km seaward of seaward-most DROT-derived measurement, we interpret this as evidence of long-term retreat. In profile 4, the MAGv2 GL is located over 5 km farther seaward than the seaward DROT-derived GL, satisfying this criterion." | 221-225 |
| 1.5 | In the phase legend of Figure 3a, it is recommended to use 180° instead of 3.14. | Fig. 3a | **Done.** | Fig. 3a |
| 1.6 | For Figures 3b–c, it is recommended to display the grounding lines derived from both methods simultaneously within the same panels, if feasible. This would enable a more direct and intuitive visual comparison between the results. | Fig. 3b-d | **Done.** | Fig. 3b-d |
| 1.7 | Comparison with Other Grounding Line Measurements: It is recommended that the authors consider incorporating a comparison with the dataset available at https://nsidc.org/data/nsidc-0778/versions/1. This product, part of the MEaSUREs program, provides a continent-wide map of short-term grounding line migration zones derived from InSAR during 2018–2020. Given its closer temporal proximity to the 2021 | | **Done.** We have incorporated a comparative analysis between our 2021 DROT-derived GZ and the MEaSUREs Antarctic GZ Version 1. This comparison provides an opportunity to assess the consistency of GZ mapping results from different techniques (DROT vs. DDInSAR) acquired within a relatively short temporal window. We now present this comparison in Section 3.2 and Figure S2, including both boundary-level analysis and spatial overlap metrics (e.g., Intersection of Union, precision, and recall). This addition helps contextualize the performance of DROT and highlights the effects of methodological and tidal difference on GZ delineation.

Line 262: "The MEaSUREs Antarctic GZ Version 1 (MAGZv1) dataset provides a comprehensive map of short-term GL migration zones across the Antarctic Ice Sheet using the DDInSAR technique (Rignot et al., 2023). We compared DROT-derived GZ results with the | 262-294; Fig. S2 |

DROT results presented in this study, it would serve as a valuable reference for contextual validation and potentially enhance the robustness of the comparative analysis.

subset of MAGZv1 dataset over the AmIS, which is based on Sentinel-1 data acquired in 2018, to assess their spatial consistency. We first computed the Intersection over Union (IoU), defined as the area of intersection divided by the area of union (Figure S2a), to evaluate the overall spatial agreement between the two GZ products. The comparison yielded an IoU of 0.44, indicating a moderate level of spatial overlap. Notably, the recall reached 0.84, suggesting that the DROT-derived GZ successfully captures the majority of the area defined by MAGZv1. However, the precision was relatively lower at 0.48, implying that over half of the area identified by DROT as GZ lies outside the extent of MAGZv1. This asymmetry reflects a broader delineation of the GZ by the DROT method, potentially capturing additional zones not included in the earlier dataset. We further evaluated the spatial offsets along the landward and seaward GZ boundaries (Figure S2b-c). For the landward boundary, the DROT-derived GZ was positioned on average 459±697 m landward relative to the MAGZv1 boundary. For the seaward boundary, the offset was -255±666 m, indicating that the DROT extend farther seaward into the floating ice shelf (Figure S2c.i and c.ii). These patterns suggest that the our DROT-derived GZ results tends to resolve a broader GZ, with boundaries shifted in opposite directions compared to MAGZv1.

We attribute the differences observed between the DROT-derived GZ and the MAGZv1 product to a combination of methodological, temporal, and tidal factors. First, the two techniques are based on fundamentally different approaches. While MAGZv1 employs the DDInSAR method to detect vertical tidal flexure through interferometric phase change, the DROT technique measures displacement from SAR amplitude imagery, enabling GZ detection even in areas with low coherence. However, DROT has a slightly lower measurement sensitivity (a fraction of a range pixel) compared to the sub-wavelength sensitivity of DDInSAR (Joughin et al., 2016). Consequently, the DROT technique tends to position the GL slightly further seaward than DDInSAR technique, consistent with our direct comparison over three representative regions, which shows a mean absolute offset of 0.35-0.42 km with standard deviations ranging from 0.14 to 0.26 km (Table 1). In addition, the two products are derived from different acquisition periods: MAGZv1 for the AmIS is based on Sentinel-1 data acquired in 2018, whereas the DROT-derived GZ uses imagery from 2021. This temporal offset means that some of the differences may reflect real GL migration over the three-year interval, though rates of change in the AmIS region are generally modest compared to dynamic West Antarctic outlets (Park et al., 2013). Lastly, both methods are sensitive to tidal conditions at the time of acquisition, but the MAGZv1 dataset does not provide metadata on tidal amplitude for each SAR acquisition. This limits our ability to directly quantify the contribution of tidal state mismatches to the observed discrepancies. In the

| | | | absence of precise tidal alignment, apparent offsets in GL position may partly reflect differences in tide-induced flexure captured at different stage of the tidal cycle. Taken together, these differences underscore the importance of method-specific sensitivities, acquisitions timing, and tidal phase alignment when comparing GL or GZ products derived from distinct remote sensing techniques." | |
|---|---|---|---|---|
| 1.8 | The manuscript states a typical offset of 1.5 km between DROT- and DDInSAR-derived grounding lines. Could the authors clarify whether this value represents an average across specific regions? Additionally, the observed differences in this study appear smaller—what factors might account for this discrepancy, and how spatially consistent are these deviations across the Amery Ice Shelf? | 304 | **Done.** We appreciate the reviewer's attention to the reported DROT–DDInSAR grounding line offset. We apologise as the previously stated value of ~1.5 km was a preliminary number. This reviewer comment prompted us to update this by performing a detailed comparison over three representative regions (b–d). The results show that the mean absolute offset between the DROT- and DDInSAR-derived grounding lines ranges from 0.35 to 0.42 km, with standard deviations of 0.14 to 0.26 km (Table 1). These values indicate that the discrepancies are smaller than our initial assessment and are spatially consistent across the AmIS within the areas of available DDInSAR measurements. We have updated the manuscript with this clarification.

Line 282: "Consequently, the DROT technique tends to position the GL slightly further seaward than DDInSAR technique, consistent with our direct comparison over three representative regions, which shows a mean absolute offset of 0.35-0.42 km with standard deviations ranging from 0.14 to 0.26 km (Table 1)." | 282-284 |
| 1.9 | The interpretation of a positive correlation between grounding line migration distance and the absolute double-differential tide range would benefit from further clarification. Does a lower absolute double-differential tide range necessarily imply smaller ice surface deformation? For instance, if the first and second SAR acquisitions occur at high tides and the third and fourth at low tides, substantial ice deformation may occur, yet the computed absolute double-differential tide range could remain small. | 355-359 | **Done.** To clarify, the absolute double-differential tidal range used in this study represents the absolute difference in ocean tide height changes between two ROT pairs. Each ROT result is derived from a SAR image pair and reflects tidal-driven ice motion at that time. While the absolute double-differential tidal range is a useful indicator of variation in tidal forcing between epochs, we acknowledge that a small double-differential tidal range does not necessarily imply low total deformation – for example, if both ROT pairs are acquired during similar tidal amplitudes (e.g., both from low-to-high tide), significant ice motion may still occur. We have added this clarification in the revised manuscript.

Line 347: "Here, the double-differential tide range refers to the absolute difference in ocean tide height change between two ROT pairs used in DROT processing. Each ROT result is computed from a SAR image pair and reflects the ice deformation induced by tidal forcing during that interval. The double-differential range therefore approximates the variation in tidal amplitude between two measurement periods. While this metric does not fully capture the cumulative tidal forcing or ice deformation. For instance, if both ROT pairs are acquired during similar tidal stages (e.g., both from low to high tide), the double-differential range may appear small, even though substantial ice deformation occurs within each pair. Despite this limitation, we observe a | 347-355 |

| | | | general trend that greater double-differential ranges are associated with broader GL migration, suggesting that short-term variability in tidal forcing plays a significant role in modulating the observed GL positions. | |
|---|---|---|---|---|
| 1.10 | Figure 2e in Chen et al. (2023) similarly demonstrates that the grounding line position of the Lambert and Mellor Glaciers oscillates between two discrete states over timescales of several months. Also, given that the temporal sampling in the present study is sparser than that of Chen et al., it is possible that some short-term transitions or episodic changes may not have been fully captured. | 375-377 | **Done.** We also agree that the lower temporal sampling in our dataset may limit the detection of short-term or transitional GL migration events. This limitation has been explicitly stated, and we now interpret the observed bistability as potentially influenced by both physical processes and observational resolution.

Line 373: "This apparent switching behaviour between discrete GL positions is consistent with previous findings (Chen et al., 2023), which documented similar bistable GL states at Lambert and Mellor Glaciers over multi-month timescales. Due to the lower temporal sampling in our study compared to Chen et al., some short-term or transitional events may not have been fully captured. As a result, the discrete nature of the observed GL states may partly reflect our sampling interval. Nevertheless, the persistence of these states across successive observations suggests a degree of stability in the GL position under specific tidal or stress conditions." | 373-378 |
| 1.11 | The synchronized grounding line migration observed for the two glaciers suggests a common driving mechanism. Could this be attributed to tidal forcing, or do the authors propose that subglacial hydrological processes—such as simultaneous subglacial lake drainage or basal melting—could exert a temporally comparable influence on both glaciers? Further discussion on the plausibility of shared controls would enhance the interpretation. | Fig. 5b | **Done**. We agree that the synchronized migration observed in profiles 14 and 15 suggests the potential influence of a shared control mechanism. In the manuscript, we examined the relationship between the GL migration and tidal forcing but found no clear correlation between the most retreated GL positions and the highest tidal amplitudes observed during the study period. This suggests that tides alone do not explain the observed behaviour.

We also discussed the possible role of subglacial hydrological processes, such as the drainage of active subglacial lakes and associated basal melting, in influencing GL migration independent of tides. We also highlight that the observed downstream shifts occurred simultaneously in both profiles, which may reflect temporally aligned subglacial water fluxes. However, due to the lack of synchronous time-series observations of subglacial lake drainage and basal melting in this region, we are currently unable to confirm this mechanism and thus discuss it only as a plausible contributing factor. | 380-419 |
| 1.12 | Is this the reason behind the classification of this island as part of the grounding zone in Figure 1c? | 455-456 | **Yes.** | 457-458 |

| 1.13 | It would be helpful to specify how the along-profile length over which the slopes are calculated was defined. What criteria were used to determine the extent of the landward and seaward segments used in the slope analysis? | 570-571 | **Done**. In our study, the slopes of the bed, surface, and ice thickness were calculated along the profile tracks by fitting the elevation data over the region between the landward and seaward bounds of the grounding zone (GZ), as shown by the black dashed lines in Figure 6e–h. These bounds correspond to the inland and seaward limits of short-term tidal migration of the grounding line, which were identified manually from the DROT results. Therefore, the length over which the slopes were computed varies between profiles, depending on the width of the GZ.

Line 572:"The slopes used in panels (b-d) were calculated by linearly fitting the respective data over the domain bounded by the black dashed lines in panels (e-h), which represent the landward and seaward limits of the tidally-induced short-term GL migration." | 572-574 |

---

## Author Comment (AC2)

**Response to reviewer #2 for 'Short and long-term grounding zone dynamics of Amery Ice Shelf, East Antarctica' - EGUSPHERE-2025-849**

Yikai Zhu, on behalf of the authors,

We thank Reviewer #2 for the insightful comments and helpful suggestions, which have contributed to improving both the scientific content and clarity of the manuscript. The table below presents a detailed response to each comment. The reviewer's comments are shown in the **Comment** column, followed by our **Response**. Changes made in the revised manuscript are referred to in the **New line** column, while the **Line** column corresponds to the original manuscript. The revised manuscript uses blue highlighting to mark all modified sections.

| ID | Comment | Line/ Figure | Response | New line |
|---|---|---|---|---|
| 2.1 | The present paper can be shortened, some detailed parts (e.g. described in the previous paper) can be moved to Supplement. Instead graphs in the Supplement (S3 – S5) which are discussed in the main paper should appear here. The novelties of the manuscript, the application of DROT method to >1100 km of AmIS boundaries, can thus be highlighted and its originality emphasized. | | **Done.** To streamline the manuscript, we have simplified the description of the inverse barometer effect (IBE) correction in Section 2.2 and moved the comparison with two grounding line datasets (previously in Section 3.2) to the Supplementary. As suggested, Supplementary Figure S3 has been relocated to the main text and now appears as revised Figure 4 to more effectively illustrate the short-term migration modes.

Figures S4 and S5, on the other hand, have been retained in the Supplementary because they present additional regional examples that support the main conclusions but are not essential to the overarching narrative. We believe keeping these figures in the Supplement helps maintain focus in the main text while still allowing interested readers to access the detailed regional analyses. | |
| 2.2 | The Grounding Line is a product of the GCOS ECV Ice Sheets and Ice Shelves and not an ECV by itself. https://gcos.wmo.int/site/global-climate-observing-system-gcos/essential-climate-variables/ice-sheets-and-ice-shelves | 23 | **Comment.** We have retained the original description of the GL as an Essential Climate Variable. As the GCOS web link provided by the reviewer shows, GL is an EVC product with associated measurement precision requirements. In our experience, the GL has always been referred to as an ECV within the context of international ESA projects, so we have retained the description as it will be understood by that relevant community. The ice sheet (or shelf) alone would not be an ECV because it is not a single measurement variable. It's possible we have misunderstood the reviewer comment, so hopefully this clarification provides useful context. | 23 |
| 2.3 | Add "atmospheric pressure" | 43 | **Done.** | 47 |
| 2.4 | "Sentinel-1A" and "Sentinel-1B" | 117-118 | **Done.** | 121-122 |

| 2.5 | The description of the IBE correction is identical to Chen et al, 2023 Section 2. This can be shortened to one sentence with citation. | 147-156 | **Done.**

Line 153: "We applied inverse barometer effect (IBE) corrections using a 1 cm/hPa conversion (Padman et al., 2003) from the fifth generation of ECMWF atmospheric reanalyses of the global climate (ERA-5) pressure anomalies, following the method described in (Chen et al., 2023)." | 153-155 |
|---|---|---|---|---|
| 2.6 | By using point H you can derive the width of the flexure zone, not the width of the GZ. The width of the GZ as far as I see in Fig 2 and explained in Section 3.1 is the range of the displacement gradient where it begins to exceed zero. | 174-175 | **Done.** Our original intent was to use the inland position where the displacement gradient increases above zero to define the GL, and the seaward location where the gradient returns to near zero as point H. The distance between these two points was used to estimate the width of the tidal flexure zone, not the grounding zone itself. We have clarified this distinction in the revised text.

Line 172: "In the seaward direction, the location where the displacement gradient approaches zero again is interpreted as point H (as defined in Figure S1)." | 172 |
| 2.7 | The GZ of MEaSUREs-Programme https://nsidc.org/data/nsidc-0778/versions/1 should be mentioned here or in Section 3.2 to refer specifically to AmIS. | 195 | **Done.** | 263 |
| 2.8 | Profile 5 seems to have a narrow GZ rather than a wide one (like profiles 14 and 15). | 201 | **Done.** We agree with the reviewer that profile 5 exhibits a relatively narrow GZ compared to other examples. To avoid confusion and better support our interpretation of wide GZ characteristics, we have removed the reference to profile 5 in the revised manuscript and retained profiles 14 and 15 as representative cases. | 200-201 |
| 2.9 | Labelling the pinning points a to k may be confused with the labels of the subplots in Figures 2 and 4 (and Supplementary S3 – S5). Maybe use brackets for the subplots e.g. (a), (b), etc. | | **Done.** We have added brackets to all subplot labels (e.g., (a), (b), etc.) to avoid confusion with the pinning point labels. For consistency, we have updated all figures in the main text and Supplementary materials to follow this unified format. | |
| 2.10 | Please be more precise what concerns the "reference lines". Are these the GLs from DDInSAR? | 246 | **Done.** We have clarified in the revised Figure 3 caption that the reference lines (green, red, and purple) are extracted from the DDInSAR-derived grounding line results.

Line 250: "These reference lines are extracted from the DDInSAR-derived GL results." | 250-251 |
| 2.11 | Correct citation (Depoorter et al, 2013b) | | **Done.** | Suppl. 87-88 |

| | | | | **Done.** In the revised manuscript, we have substantially shortened Section 3.2 to improve clarity and focus. Specifically, we removed the comparison with the Synthesized GL and MAGv2 datasets from the main text and moved the corresponding table and figures to the Supplementary Material, as these products are based on much earlier observations and are less directly comparable to the 2021 DROT-derived GLs. | |
|---|---|---|---|---|---|
| 2.12 | | Section 3.2 is very long. A comparison between DROT and DInSAR was already shown in (Wallis et al, 2024). Starting with line 257 the comparison with the two published datasets (text and Table 2) may be moved to the Supplement or removed. As already mentioned in the paper these datasets do not have the same time stamp as the DROT GLs since they are based on at least 2 decades older data. The discussion on the bias between DDInSAR and DROT GL position should focus on the 2021 datasets. As mentioned above I suggest also to add here the comparison to the MEaSUREs GZ on AmIS. | | In accordance with the reviewer's recommendation, we now focus our discussion on the comparison between DROT and contemporaneous DDInSAR-derived GLs from 2021, which provides a more direct and robust validation of the DROT technique. Additionally, we have included a new comparison with the MEaSUREs Antarctic Grounding Zone Version 1 (MAGZv1) dataset over the Amery Ice Shelf, which is derived from Sentinel-1 data acquired in 2018. Although not perfectly contemporaneous, this product offers valuable spatial context for assessing the consistency of grounding zone mapping across techniques. The results of this comparison are now presented in Section 3.2 and Figure S2.

Lines 262:" The MEaSUREs Antarctic GZ Version 1 (MAGZv1) dataset provides a comprehensive map of short-term GL migration zones across the Antarctic Ice Sheet using the DDInSAR technique (Rignot et al., 2023). We compared DROT-derived GZ results with the subset of MAGZv1 dataset over the AmIS, which is based on Sentinel-1 data acquired in 2018, to assess their spatial consistency. We first computed the Intersection over Union (IoU), defined as the area of intersection divided by the area of union (Figure S2a), to evaluate the overall spatial agreement between the two GZ products. The comparison yielded an IoU of 0.44, indicating a moderate level of spatial overlap. Notably, the recall reached 0.84, suggesting that the DROT-derived GZ successfully captures the majority of the area defined by MAGZv1. However, the precision was relatively lower at 0.48, implying that over half of the area identified by DROT as GZ lies outside the extent of MAGZv1. This asymmetry reflects a broader delineation of the GZ by the DROT method, potentially capturing additional zones not included in the earlier dataset. We further evaluated the spatial offsets along the landward and seaward GZ boundaries (Figure S2b-c). For the landward boundary, the DROT-derived GZ was positioned on average 459±697 m landward relative to the MAGZv1 boundary. For the seaward boundary, the offset was -255±666 m, indicating that the DROT extend farther seaward | 262-294 |

| | | | into the floating ice shelf (Figure S2c.i and c.ii). These patterns suggest that the our DROT-derived GZ results tends to resolve a broader GZ, with boundaries shifted in opposite directions compared to MAGZv1.

We attribute the differences observed between the DROT-derived GZ and the MAGZv1 product to a combination of methodological, temporal, and tidal factors. First, the two techniques are based on fundamentally different approaches. While MAGZv1 employs the DDInSAR method to detect vertical tidal flexure through interferometric phase change, the DROT technique measures displacement from SAR amplitude imagery, enabling GZ detection even in areas with low coherence. However, DROT has a slightly lower measurement sensitivity (a fraction of a range pixel) compared to the sub-wavelength sensitivity of DDInSAR (Joughin et al., 2016). Consequently, the DROT technique tends to position the GL slightly further seaward than DDInSAR technique, consistent with our direct comparison over three representative regions, which shows a mean absolute offset of 0.35-0.42 km with standard deviations ranging from 0.14 to 0.26 km (Table 1). In addition, the two products are derived from different acquisition periods: MAGZv1 for the AmIS is based on Sentinel-1 data acquired in 2018, whereas the DROT-derived GZ uses imagery from 2021. This temporal offset means that some of the differences may reflect real GL migration over the three-year interval, though rates of change in the AmIS region are generally modest compared to dynamic West Antarctic outlets (Park et al., 2013). Lastly, both methods are sensitive to tidal conditions at the time of acquisition, but the MAGZv1 dataset does not provide metadata on tidal amplitude for each SAR acquisition. This limits our ability to directly quantify the contribution of tidal state mismatches to the observed discrepancies. In the absence of precise tidal alignment, apparent offsets in GL position may partly reflect differences in tide-induced flexure captured at different stage of the tidal cycle. Taken together, these differences underscore the importance of method-specific sensitivities, acquisitions timing, and tidal phase alignment when comparing GL or GZ products derived from distinct remote sensing techniques." | |
| 2.13 | Swap *ocean* and *land*. "landward GL" … refer to the GLs closest to the land. | 291 | **Done.** | Suppl. 46 |

| | | | | |
|---|---|---|---|---|
| 2.14 | Figure 4 and Figure S3 are almost identical. I suggest to add all profiles (those with no clear mode as well) to Figure 4 in the main text where you discuss all patterns (and delete Figure S3 in the supplement). Figure 4 and Figure S3: how is the "0" of the Y-axis defined? | 344-345 | **Done.** Figure S3 has been moved to the main text as part of the revised Figure 4, which now includes all profiles. We have also clarified the meaning of the Y-axis zero in the new Figure 4 caption: Note that a GL migration distance of 0 km represents the location of the seawardmost GL observed in each profile, which is used as the reference point for calculating relative migration distances. | 324-334 |
| 2.15 | Figure S4; Figure 4l $R^2$=0.83 while in Figure S4q $R^2$=0.84; I suggest to mention that the profiles 7,8, and 17 were selected due to their different migration pattern. | | The discrepancy between the $R^2$ values in Figure 4l and Figure S4q was due to a typographical error — both should be 0.84, and this has now been corrected. Since the revised Figure 4 now focuses solely on short-term migration, we did not include this figure in the main text. Regarding the suggestion to mention that profiles 7, 8, and 17 were selected due to their different migration patterns. We would appreciate further clarification regarding the intended distinction. | |
| 2.16 | … it then transits downstream | 371 | **Done.** | 368 |
| 2.17 | Correct reference for Freer et al, 2023. Freer, B. I. D., Marsh, O. J., Hogg, A. E., Fricker, H. A., and Padman, L.: Modes of Antarctic tidal grounding line migration revealed by Ice, Cloud, and land Elevation Satellite-2 (ICESat-2) laser altimetry, The Cryosphere, 17, 4079–4101, https://doi.org/10.5194/tc-17-4079-2023, 2023. | 665 | **Done.** | 668-669 |
| 2.18 | Figure S1 (b): at high tide the GL moves landward, therefore (Fmax, Gmax) correspond to low tide | Fig. S1 | **Done.** | Fig. S1 |

---

## Author Response (AR2)

**Response to editor for 'Short and long-term grounding zone dynamics of Amery Ice Shelf, East Antarctica' - EGUSPHERE-2025-849**

Yikai Zhu, on behalf of the authors,

We sincerely thank the editor for the thorough review of our revised manuscript and for confirming that the referees' comments have been adequately addressed. We also greatly appreciate the additional detailed feedback provided, which has helped us further improve the clarity and precision of the manuscript. Below, we provide a point-by-point response to each of the editor's comments. Editor's comments are reproduced in the **Comment** column, with our **Response** listed alongside. The **Line** column refers to the position of the relevant text in the original submission, and the **New line** column indicates where changes were made in the revised version. All modified text is highlighted in blue to clearly indicate revisions made in response to editor's feedback.

| ID | Comment | Line | Response | New line |
|---|---|---|---|---|
| 1 | The word "amplitude" is used to as if it meant "tidal displacement." The two are different- the amplitude is the peak displacement of a periodic signal relative to an equilibrium point- so the tidal amplitude represents the maximum of the tidal displacement relative to some standard value and is always positive. The manuscript should use "tidal displacement" almost everywhere the term "tidal amplitude" is used. | | **Done.** We have revised the manuscript accordingly, replacing "tidal amplitude" with "tidal displacement" throughout the text. In addition, all relevant figure labels and annotations have been updated to reflect this correction, including those in Figure 2s, Figure 4 and Figure S5. | |
| 2 | please provide a citation for the MeaSUREs dataset | 16 | **Done.** | 16 |
| 3 | mischaracterized can be a single word (no hyphen is appropriate) | 53 | **Done.** | 53 |
| 4 | "are likely to" is probably an overstatement- "may" or "might" is more appropriate | 63 | **Done.** We have replaced "are likely to" with "may". | 64 |
| 5 | specify "East Antarctic coastline"—the peninsula extends much farther north | 95 | **Done.**

"AmIS is distinct from other Antarctic ice shelves due to its long, narrow sub-ice-shelf cavity and its location at the northernmost latitude (69°S) of the **East** Antarctic | 96 |

| | | | coastline, which exposes it to different environmental conditions." | |
|---|---|---|---|---|
| 6 | should be "low rates of basal melt" | 96 | **Done.** "Unlike ice shelves in the Amundsen Sea and on the Antarctic Peninsula, AmIS has relatively **low rates of basal melt**, with modest thickening across the majority of its area and thinning at its most inland reaches" | 97 |
| 7 | "the identical satellite pass" is not clear. "a satellite pass with approximately identical geometry" would be better, particularly if an approximate value were provided for how identical the geometry needs to be. | 159 | **Done.** "we therefore difference two ROT displacement fields derived from **satellite passes with approximately identical imaging geometry**" | 159-160 |
| 8 | The graphic in figure 1 (and, less clearly, some of the text in 3.4.3) suggests that the entirety of Gillock island experiences tidal displacement. The center of the island has a surface height that is well above flotation, which makes it difficult to see how this could be so. Likewise, panel 1(f) seems to show little or no displacement over the top of the island. What am I missing here? | Figure 1 | **Done.** We agree that the previous version of Figure 1(c) may have incorrectly suggested that tidal displacement – and hence the GZ – extended across the entirety of Gillock Island. This misinterpretation was due to an overestimation of GZ extent in this region, which erroneously included the central, grounded part of the island. In the revised figure, we have corrected this by refining the GZ delineation near Gillock Island and excluded the central part of the island from the GZ. This change better reflects the true spatial extent of tidal response and avoids the previous implication that the island is fully within the GZ. | Figure 1 |
| 9 | This paragraph belongs in the next section. | 218-224 | **Done.** We have moved this paragraph to the end of section 3.2. | 293-299 |

| 10 | The MAGv2 dataset has not been defined. Is this the same as the MeaSURES grounding line defined in the next section? | 223 | **Done.** We have now provided the full definition of the MAGv2 dataset as MEaSUREs Antarctic GL Version2. To avoid confusion with MEaSUREs Antarctic GZ Version1 (MAGZv1) dataset mentioned in the following section, we have renamed MAGv2 to MAGLv2 throughout the text.

"we compare our DROT-derived seaward GL position ($F_{max}$) with the historical GL location from **MEaSUREs Antarctic GL Version2 (MAGLv2)** dataset" | 297 |
|---|---|---|---|---|
| 11 | "is shown to be precise" It would be better to define precise here: for example "is shown to be precise to less than 0.5 km" | 242 | **Done.**

"The DROT-derived GL's alignment with the DDInSAR measurements is shown to be precise **within 0.5 km, with differences** ranging from 0.35±0.14 km to 0.42±0.26 km" | 236 |
| 12 | The expression "mean absolute" of quantity x often means "mean(\|x\|)" and is used as a measure of the spread of a distribution. I think in this case, it just means mean(x). Am I correct? If so, please remove the term "absolute". I would suggest using "mean horizontal separation" or (maybe better) "mean seaward separation" | Table 1 | **Done.** You are correct – in our case, the values represent the mean separation between DROT- and DDInSAR-derived GLs, and are not absolute values in the sense of mean(\|x\|). We have therefore removed the word "absolute" to avoid confusion, and revised the term to "mean seaward separation" throughout the text. | Table 1 |
| 13 | Please explain what is meant by "the recall reached 0.84" | 266 | **Done.** We have added a brief explanation of the recall and precision metrics to the main text.

"**In addition, we calculated recall and precision to quantify detection performance. Here, recall is defined as the fraction of the MAGZv1 GZ area that is correctly identified by the DROT-derived GZ, while precision refers to the fraction of the DROT-derived GZ area that overlaps with MAGZv1.**" | 261-263 |
| 14 | A quantity on the order of 200-400 m with an uncertainty of ~700 m should have only one significant figure, but two are sometimes permitted. Please change to 460±700 m and -260±760 m. | 271-273 | **Done.**

"For the landward boundary, the DROT-derived GZ was positioned on average **460±700** m landward relative to the MAGZv1 boundary. For the seaward boundary, the offset was **-260±660** m, indicating that the DROT extend farther seaward into the floating ice shelf (Figure S2c.i and c.ii)." | 268-269 |

| 15 | "with boundaries shifted in opposite directions compared to MAGZv1" – This is not very clear- I'd either delete it, or restate that the landward boundary was shifted landward and the seaward boundary was shifted seaward. | 274 | **Done.**

"These patterns suggest that our DROT-derived GZ results tends to resolve a broader GZ, **with landward boundary positioned farther inland and the seaward boundary extending farther into the floating ice shelf** compared to MAGZv1." | 271-272 |
|---|---|---|---|---|
| 16 | Please provide units for a "fraction of a range pixel" and "sub-wavelength" for Sentinel-1. | 280 | **Done.**

"DROT has a slightly lower measurement sensitivity, **typically on the order of 0.2-0.7 m,** corresponding to a fraction of the ~2.3 m range pixel in Sentinel-1 images, compared to the sub-wavelength sensitivity **(~1-10 mm)** of DDInSAR" | 288-289 |
| 17 | See my prior comment on "mean absolute offset" | 283 | **Done.** | 281 |
| 18 | Please rewrite with the study name in parentheses (e.g. "we adopt the categorization established in a previous study (Freer et al, 2023)). | 297 | **Done.**

"we adopt the categorization framework established in **a previous study** (Freer et al., 2023)" | 302 |
| 19 | no hyphen between active and lakes | 380 | **Done.**

"A chain of **active lakes** was discovered beneath Lambert Glacier" | 385 |

---

## Author Response (AR3)

**Response to editor for 'Short and long-term grounding zone dynamics of Amery Ice Shelf, East Antarctica' - EGUSPHERE-2025-849**

Yikai Zhu, on behalf of the authors,

We sincerely thank the editor for taking the time to handle our revised manuscript even during your vacation. We greatly appreciate your careful attention to detail and your constructive suggestions throughout the review process. In accordance with your comment, we have now standardized the terminology related to tides across the manuscript. Specifically, when referring to the tidal value at a given moment, we now consistently use tidal height.